# Variational Learning ISTA

## Abstract

Compressed sensing combines the power of convex optimization techniques with a sparsity inducing prior on the signal space to solve an underdetermined system of equations. For many problems, the sparsifying dictionary is not directly given, nor its existence can be assumed. Besides, the sensing matrix can change across different scenarios. Addressing these issues requires solving a sparse representation learning problem, namely dictionary learning, taking into account the epistemic uncertainty on the learned dictionaries and, finally, jointly learning sparse representations and reconstructions under varying sensing matrix conditions. We propose a variant of the LISTA architecture that incorporates the sensing matrix into the architecture. In particular, we propose to learn a distribution over dictionaries via a variational approach, dubbed Variational Learning ISTA (VLISTA), which approximates a posterior distribution over the dictionaries as part of an unfolded LISTA-based recovery network. Such a variational posterior distribution is updated after each iteration, and thereby adapts the dictionary according to the optimization dynamics. As a result, VLISTA provides a probabilistic way to jointly learn the dictionary distribution and the reconstruction algorithm with varying sensing matrices. We provide theoretical and experimental support for our architecture and show that it learns calibrated uncertainties.

## 1   Introduction

Compressed sensing methods aim at solving under-determined inverse problems imposing a prior about signal structure. Sparsity and linear inverse problems were canonical examples of the signal structure and sensing mediums (modelled with a linear transformation $\Phi$). Many works during recent years focused on improving the performance and complexity of compressed sensing solvers for a given dataset. A typical approach is based on unfolding iterative algorithms as layers of neural networks and learning the parameters end-to-end starting from learning iterative soft thresholding algorithm (LISTA) Gregor & LeCun (2010) with many follow-ups works. Varying sensing matrices and unknown sparsifying dictionaries are some of the main challenges of data-driven approaches. The works in Aberdam et al. (2021); Schnoor et al. (2022) address these issues by learning a dictionary and include it in the optimization iteration. However, the data samples might not have any exact sparse representations, which means that there is no ground truth dictionary. The issue can be more severe for heterogeneous datasets where the choice of the dictionary might vary from one sample to another. A principled approach to this problem would be to take a Bayesian approach and define a distribution over the learned dictionaries with proper uncertainty quantification.

In this work, first, we formulate an augmented LISTA-like model, termed Augmented Dictionary Learning ISTA (A-DLISTA), that can adapt its parameters to the current data instance. We theoretically motivate such a design and empirically prove that it can outperform other LISTA-like models in a non-static measurement scenario, i.e., considering varying sensing matrices across data samples. We are aware that an augmented version of LISTA, named Neurally Augmented ALISTA (NALISTA), was already proposed in Behrens et al. (2021), however, there are some fundamental differences between NALISTA and A-DLISTA. First, our model takes as input the per-sample sensing matrix and the dictionary at the current layer. This means that A-DLISTA adapts the parameters to the

Preprint. Under review.

current measurement setup as well as to the learned dictionaries. In contrast, NALISTA assumes to have a fixed sensing matrix to analytically evaluate its weight matrix, $\mathbf{W}$. Hypothetically, NALISTA could handle varying sensing matrices, however, that comes at the price of having to solve for each data sample the inner optimization step to evaluate the $\mathbf{W}$ matrix. Moreover, the architectures of the augmentation networks are profoundly different. Indeed, while NALISTA uses an LSTM, A-DLISTA employ a convolutional neural network, shared across all layers. Such a different choice reflects the different types of dependencies between layers and input data that the networks try to model. We report in subsection 3.3 a detailed discussion about the theoretical motivation and architectural design for A-DLISTA. Moreover, the detailed architecture is described in Appendix A.

Finally, we introduce Variational Learning ISTA (VLISTA) where we learn a distribution over dictionaries and update it after each iteration based on the outcome of the previous layer. In this sense, our model learns an adaptive iterative optimization algorithm where the dictionary is iteratively refined for the best performance. Besides, the uncertainties estimation provides an indicator for detecting Out-Of-Distribution (OOD) samples. Intuitively, our model can be understood as a form of a recurrent variational autoencoder, e.g., Chung et al. (2015), where on each iteration of the optimization algorithm, we have an approximate posterior distribution over the dictionaries, conditioned on the outcome of the last iteration. The main contributions of our work are as follows.

- We design an augmented version of LISTA, dubbed A-DLISTA, that can handle non-static measurement setups, i.e., per-sample sensing matrices, and that can adapt parameters to the current data instance.
- We propose Variational Learning ISTA (VLISTA) that learns a distribution over sparsifying dictionaries. The model can be interpreted as a Bayesian LISTA model that leverage A-DLISTA as the likelihood model.
- VLISTA adapts the dictionary to optimization dynamics and therefore can be interpreted as a hierarchical representation learning approach, where the dictionary atoms gradually permit more refined signal recovery.
- The dictionary distributions can be used for out-of-distribution detection.

The remaining part of the paper is organized as follows. In section 2 we briefly report related works that are relevant to the current research, while in section 3 the model formulation is detailed. The datasets description, as well as the experimental results, are reported in section 4. Finally, we report our conclusion in section 5.

## 2   Related Works

Compressed sensing field is abound with works on theoretical and numerical analysis of recovery algorithms (see Foucart & Rauhut (2013)) with iterative algorithms as one of the central approaches like Iterative Soft-Thresholding Algorithm (ISTA) Daubechies et al. (2004), Approximate message passing (AMP) Donoho et al. (2009) Orthogonal Matching Pursuit (OMP) Pati et al. (1993); Davis et al. (1994) and Iterative Hard-Thresholding Algorithm (IHTA) Blumensath & Davies (2009). The mentioned algorithms are characterized by a specific set of hyperparameters, e.g., number of iterations and soft thresholds, that can be tuned to obtain a better trade-off between performance and complexity. With unfolding iterative algorithms as layers of neural networks, these parameters can be learned in an end-to-end fashion from a dataset, see for instance some variants Zhang & Ghanem (2018); Metzler et al. (2017); yang et al. (2016); Borgerding et al. (2017); Sprechmann et al. (2015).

**Bayesian Compressed Sensing (BCS) and Dictionary learning.**  A non-parametric Bayesian approach to dictionary learning has been introduced in Zhou et al. (2009, 2012), where the authors consider a fully Bayesian joint compressed sensing inversion and dictionary learning. Besides, their atoms are drawn and fixed a priori. Bayesian compressed sensing Ji et al. (2008) leverages relevance vector machines (RVMs) Tipping (2001) and uses a hierarchical prior to model distributions of each entry. This line of work quantifies uncertainty of recovered entries while assuming a fixed dictionary. In contrast, in our work, the source of uncertainty is the unknown dictionary over which we define a distribution.

**LISTA models.** Learning ISTA was first introduced in Gregor & LeCun (2010) with many follow-up variations. The follow-up works in Behrens et al. (2021); Liu et al. (2019); Chen et al. (2021); Wu

et al. (2020) provides various guidelines for architecture change to improve LISTA for example in convergence, parameter efficiency, step size and threshold adaptation, and overshooting. The common assumptions of these works are fixed and known sparsifying dictionary and fixed sensing matrix. Steps toward relaxing these assumptions were taken in Aberdam et al. (2021); Behboodi et al. (2022); Schnoor et al. (2022). In Aberdam et al. (2021), the authors propose a model to deal with varying sensing matrix (dictionary). The authors in Schnoor et al. (2022); Behboodi et al. (2022) provide an architecture that can both incorporate varying sensing matrices and learn dictionaries, although their focus is on learning theoretical analysis of the model. There are theoretical studies on the convergence and generalization of unfolded networks, see for example Giryes et al. (2018); Pu et al. (2022); Aberdam et al. (2021); Pu et al. (2022); Chen et al. (2018); Behboodi et al. (2022); Schnoor et al. (2022). In our paper, not only we consider varying sensing matrix and dictionary, but we also model distribution over dictionaries and thereby characterizing the epistemic uncertainty.

**Recurrent Variational models.** Variational autoencoders (VAEs) is a framework, that learns a generative model over the data through latent variables Kingma & Welling (2013); Rezende et al. (2014). When there are data-sample specific dictionaries in our proposed model, it reminisces extensions of VAEs to the recurrent setting Chung et al. (2015, 2016), which assumes a sequential structure in the data and imposes temporal correlations between the latent variables. There are also connections and similarities to Markov state-space models, such as the ones described at Krishnan et al. (2017).

**Bayesian Deep Learning.** When we employ global dictionaries in VLISTA, the model essentially becomes a variational Bayesian Recurrent Neural Network. Variational Bayesian neural networks have been introduced at Blundell et al. (2015), with independent priors and variational posteriors for each layer. This work has been further extended to recurrent settings at Fortunato et al. (2019). The main difference between these works with our setting is the prior and variational posterior; in our case where the prior and variational posterior for each step is conditioned on previous steps, instead of being fixed across steps.

## 3 Variational Learning ISTA

In this section, we briefly report on the ISTA and LISTA models to solve linear inverse problems. Then, we introduce our first model, A-DLISTA, capable of learning the sparsifying dictionary and adapting to different sensing matrices. Finally, we focus on the VISTA model, a variational framework for solving linear inverse problems that leverages A-DLISTA as the likelihood model and achieves high power to reject OOD samples.

### 3.1 Linear inverse problems

We consider the following linear inverse problem: $\boldsymbol{y} = \boldsymbol{\Phi}\boldsymbol{x}$.

The matrix $\boldsymbol{\Phi}$ is called the sensing matrix. If the vector $\boldsymbol{x}$ is sparse in a dictionary basis $\boldsymbol{\Psi}$, the problem can be cast as a sparse recovery problem $\boldsymbol{y} = \boldsymbol{\Phi}\boldsymbol{\Psi}\boldsymbol{z}$ with $\boldsymbol{z}$ given as a sparse vector. A proximal gradient descent-based approach to this problem yields ISTA iterations:

$$\boldsymbol{z}_t = \eta_{\theta_t}\left(\boldsymbol{z}_{t-1} + \gamma_t(\boldsymbol{\Phi}\boldsymbol{\Psi})^H(\boldsymbol{y} - \boldsymbol{\Phi}\boldsymbol{\Psi}\boldsymbol{z}_{t-1})\right), \tag{1}$$

where $\theta_t, \gamma_t > 0$ are hyper-parameters of the model meaning that the algorithm does not possess any trainable parameters. Generally speaking, $\gamma_t$ is called the step size and its value is given as the inverse of the spectral norm of the matrix $\boldsymbol{A}$, where $\boldsymbol{A} = \boldsymbol{\Phi}\boldsymbol{\Psi}$. The hyper-parameter $\theta_t$ is termed threshold and it is the value characterizing the so-called soft-threshold function given by: $\eta_{\theta}(\boldsymbol{x}) = \text{sign}(\boldsymbol{x})(|\boldsymbol{x}| - \theta)_+$. In the ISTA formulation, those two parameters are shared across all the iterations. Therefore, we have $\gamma_t, \theta_t \rightarrow \gamma, \theta$.

### 3.2 LISTA

LISTA Gregor & LeCun (2010) is a reparametrized unfolded version of the ISTA algorithm in which each iteration, or layer, is characterized by learnable matrices. Specifically, LISTA reinterpret Equation 1 as defining the layer of a feed-forward neural network implemented as $S_{\theta_t}\left(\boldsymbol{V}_t\boldsymbol{x}_{t-1} + \boldsymbol{W}_t\boldsymbol{y}\right)$ where $\boldsymbol{V}_t, \boldsymbol{W}_t$ are learnt from a dataset. In that way, those weights implicitly contain information

about $\mathbf{\Phi}$ and $\mathbf{\Psi}$. However, in many problems, the dictionary $\mathbf{\Psi}$ is not given, and the sensing matrix $\mathbf{\Phi}$ can change for each sample in the dataset. As LISTA, also its variations, e.g., Analytic LISTA (ALISTA) Liu et al. (2019), NALISTA Behrens et al. (2021) and HyperLISTA Chen et al. (2021), require similar constraints such a fix dictionary and sensing matrix. Thus, making those algorithms fail in situations where either $\mathbf{\Phi}$ is not fixed or $\mathbf{\Psi}$ is not known.

### 3.3  Augmented Dictionary Learning ISTA

To deal with situations where the underlying dictionary is not known, and moreover the sensing matrix is changing across samples, one can use an unfolded version of ISTA in which the dictionary is considered as a learnable matrix, termed Dictionary Learning ISTA (DLISTA), for which each layer is given as follows:

$$z_t = \eta_{\theta_t} \left( z_{t-1} + \gamma_t (\mathbf{\Phi}\mathbf{\Psi}_t)^\top (y - \mathbf{\Phi}\mathbf{\Psi}_t z_{t-1}) \right), \tag{2}$$

with one last linear layer mapping $z$ to reconstructed input. The model can be trained end to end to learn all $\theta_t, \gamma_t, \mathbf{\Psi}_t$. The base model is very similar to Behboodi et al. (2022); Aberdam et al. (2021) but as we will see further, it requires additional changes. To see this, consider the basic scenario where the sensing matrix is fixed to $\mathbf{\Phi}$, there is a ground-truth (unknown) dictionary $\mathbf{\Psi}_o$ such that $x_* = \mathbf{\Psi}_o z_*$ with sparse $z_*$ having support $S$, i.e., $\text{supp}(z_*) = S$.

Consider the layer $t$ of DLISTA with a fixed sensing matrix $\mathbf{\Phi}$, and define:

$$\tilde{\mu} := \max_{1 \leq i \neq j \leq N} \left| ((\mathbf{\Phi}\mathbf{\Psi}_t)_i)^\top (\mathbf{\Phi}\mathbf{\Psi}_t)_j \right| \tag{3}$$

$$\tilde{\mu}_2 := \max_{1 \leq i,j \leq N} \left| ((\mathbf{\Phi}\mathbf{\Psi}_t)_i)^\top (\mathbf{\Phi}(\mathbf{\Psi}_t - \mathbf{\Psi}_o))_j \right| \tag{4}$$

$$\delta(\gamma) := \max_i \left| 1 - \gamma \left\| (\mathbf{\Phi}\mathbf{\Psi}_t)_i \right\|_2^2 \right| \tag{5}$$

The term $\tilde{\mu}$ is called the mutual coherence of the matrix $\mathbf{\Phi}\mathbf{\Psi}_t$. The term $\tilde{\mu}_2$ is closely connected to generalized mutual coherence, however it differs in that unlike generalized mutual coherence, it includes the diagonal inner product for $i = j$. Finally, the term $\delta(\gamma)$ is the reminiscent of restricted isometry property (RIP) constant Foucart & Rauhut (2013), a key condition for many recovery guarantees in compressed sensing. Note that there is a dependency on $\gamma$. For simplicity, we only kept the dependence on $\gamma$ in the notation and dropped the dependence of $\tilde{\mu}, \tilde{\mu}_2$ and $\delta$ on $\mathbf{\Phi}$ and $\mathbf{\Psi}_t$ from the notation.

The following theorem provides conditions on each layer improving the reconstruction error.

**Theorem 3.1.** *Consider the layer $t$ of DLISTA given by equation 2, and suppose that $y = \mathbf{\Phi}\mathbf{\Psi}_o z_*$ with $\text{supp}(z_*) = S$. We have*

1. *Suppose $z_{t-1}$ has the same support as $z_*$, i.e., $\text{supp}(z_*) = \text{supp}(z_{t-1})$. If*

$$\gamma_t \left( \tilde{\mu} \left\| z_* - z_{t-1} \right\|_1 + \tilde{\mu}_2 \left\| z_* \right\|_1 \right) \leq \theta_t, \tag{6}$$

*then $\text{supp}(z_t) \subseteq \text{supp}(z_*)$.*

2. *Assuming that the conditions of the last step hold, then we get the following bound on the error:*

$$\left\| z_t - z_* \right\|_1 \leq (\delta(\gamma_t) + \gamma_t \tilde{\mu}(|S| - 1)) \left\| z_{t-1} - z_* \right\|_1 + \gamma_t \tilde{\mu}_2 |S| \left\| z_* \right\|_1 + |S| \theta_t.$$

We provide the derivations in the supplementary materials. Theorem 3.1 provides insights about the choice of $\gamma_t$ and $\theta_t$, and also suggests that $(\delta(\gamma_t) + \gamma_t \tilde{\mu}(|S| - 1))$ needs to be smaller than one to reduce the error at each step. Similar to many existing works in the literature, Theorem 3.1 emphasizes the role of small mutual coherence, equation 3, for good convergence.

Looking at the theorem, it can be seen that $\gamma_t$ and $\theta_t$ play crucial role for the convergence. However, there is trade-off underlying these choices. Let's fix $\theta_t$. Decreasing $\gamma_t$ can guarantee good support selection but can increase $\delta(\gamma_t)$. When the sensing matrix is fixed, the network can hopefully find good choices by end-to-end training. However, when the sensing matrix $\mathbf{\Phi}$ changes across different data samples, i.e., $\mathbf{\Phi} \rightarrow \mathbf{\Phi}^i$, it is not guaranteed anymore that there is a unique choice of $\gamma_t$ and $\theta_t$ for all different $\mathbf{\Phi}^i$. Since these parameters can be determined for a fixed $\mathbf{\Phi}$ and $\mathbf{\Psi}_t$, we propose using an augmentation network that determines $\gamma_t$ and $\theta_t$ from each pair of $\mathbf{\Phi}$ and $\mathbf{\Psi}_t$. Following from theory, we show in Figure 1 the resulting model named A-DLISTA.

A-DLISTA relies on two basic operations at each layer, namely, soft-threshold (blue blocks in Figure 1) and augmentation (red blocks in Figure 1). The former represents an ISTA-like iteration parametrized by the set of learnable weights: $\{\boldsymbol{\Psi}_t, \theta_t, \gamma_t\}$, whilst the latter is implemented using an encoder-decoder-like type of network. As shown in the figure, the augmentation network takes as input the sensing matrix for the given data sample, $\boldsymbol{\Phi}^i$, together with the dictionary learned at the layer for which the augmentation model will generate the $\theta$ and $\gamma$ parameters. Through such an operation, the A-DLISTA adapts those last two parameters to the current data sample. We report more details about the augmentation network in Appendix A.

## 3.4 VLISTA

Although A-DLISTA possesses adaptivity to data samples, it is still based on the assumption that a ground truth dictionary exists. We relax that hypothesis by defining a probability distribution over the sparsifying dictionary and formulate a variational approach, titled VLISTA,

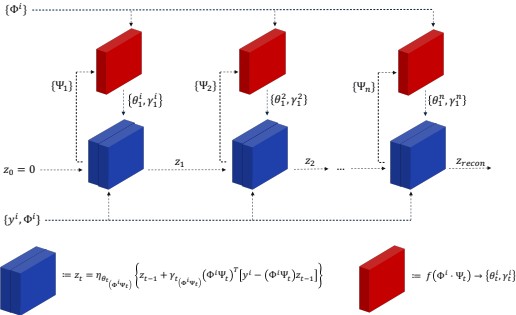

Figure 1: A-DLISTA architecture. The blue blocks represent a single soft-thresholding operation parametrized by the dictionary $\boldsymbol{\Psi}_t$ together with threshold and step size $\{\theta_t, \gamma_t\}$ at layer $t$. The red blocks represent the augmentation network (with shared parameters across layers) that adapts $\{\theta_t, \gamma_t\}$ for layer $t$ based on the dictionary $\boldsymbol{\Psi}_t$ and the current measurement setup $\boldsymbol{\Phi}^i$ for the $i-$th data sample. The dashed arrows connecting each blue block with a red one mean that, at each iteration, the augmentation newtork receives the learned dictionary at the current iteration as input (together with the sensing matrix).

to jointly solve the dictionary learning and the sparse recovery problems. To forge our variational framework whilst retaining the helpful adaptivity property of A-DLISTA, we re-interpret the soft-thresholding layers of the latter as part of a likelihood model that defines the output mean for the reconstructed signal. Given its recurrent-like structure Chung et al. (2015), we equip VLISTA with a conditional trainable prior where the condition is given by the dictionary sampled at the previous iteration. Therefore, the full model comprises three components, namely, the conditional prior $p_\xi(\cdot)$, the variational posterior $q_\phi(\cdot)$, and the likelihood model, $p_\Theta(\cdot)$. All components are parametrized by neural networks whose outputs represent the parameters for the underlying probability distribution. In what follows, we describe more in detail the various building blocks of the VLISTA model.

### 3.4.1 Prior distribution over dictionaries

The conditional prior, $p_\xi(\boldsymbol{\Psi}_t|\boldsymbol{\Psi}_{t-1})$, is modelled as a Gaussian distribution with parameters conditioned on the previously sampled dictionary. We parametrize $p_\xi(\cdot)$ using a neural network, $f_\xi(\cdot) = [f_{\xi_1}^\mu \circ g_{\xi_0}(\cdot), f_{\xi_2}^{\sigma^2} \circ g_{\xi_0}(\cdot)]$, with trainable parameters $\xi = \{\xi_0, \xi_1, \xi_2\}$. The model's architecture comprises a shared convolutional block followed by two different branches generating the mean and the standard deviation, respectively, of the Gaussian distribution. Therefore, at layer $t$, the prior conditional distribution is given by: $p_\xi(\boldsymbol{\Psi}_t|\boldsymbol{\Psi}_{t-1}) = \prod_{i,j} \mathcal{N}(\boldsymbol{\Psi}_{t,i,j}|\mu_{t,i,j} = f_{\xi_1}^\mu(g_{\xi_0}(\boldsymbol{\Psi}_{t-1}))_{i,j}; \sigma_{t,i,j} = f_{\xi_2}^{\sigma^2}(g_{\xi_0}(\boldsymbol{\Psi}_{t-1}))_{i,j})$, where the indices $i,j$ run over the rows and columns of $\boldsymbol{\Psi}_t$. In order to simplify our expressions, we will abuse notation and refer to distributions like the former as $p_\xi(\boldsymbol{\Psi}_t|\boldsymbol{\Psi}_{t-1}) = \mathcal{N}(\boldsymbol{\Psi}_t|\boldsymbol{\mu}_t = f_{\xi_1}^\mu(g_{\xi_0}(\boldsymbol{\Psi}_{t-1})); \boldsymbol{\sigma}^2_t = f_{\xi_2}^{\sigma^2}(g_{\xi_0}(\boldsymbol{\Psi}_{t-1})))$. We will use the same type of notation throughout the rest of the manuscript to simplify formulas. The prior's design allows for enforcing a dependence of the dictionary at iteration $t$ to the one sampled at the previous iteration. Thus, allowing us to refine the $\boldsymbol{\Psi}$ as the iterations proceed. The only exception to such a process is the prior imposed over the dictionary at $t = 1$, since there is no a previously sampled dictionary in this case. We handle such an exception by assuming a standard Gaussian distributed $\boldsymbol{\Psi}_1$. Finally, the joint prior distribution over the dictionaries for VLISTA is given by:

$$p_\xi(\boldsymbol{\Psi}_{1:T}) = \mathcal{N}(\boldsymbol{\Psi}_1|\boldsymbol{\mu} = \mathbf{0}; \boldsymbol{\sigma}^2 = \mathbf{1}) \prod_{t=2}^{T} \mathcal{N}(\boldsymbol{\Psi}_t|\boldsymbol{\mu}_t = f_{\xi_1}^{\boldsymbol{\mu}}(g_{\xi_0}(\boldsymbol{\Psi}_{t-1})); \boldsymbol{\sigma}^2_t = f_{\xi_2}^{\sigma^2}(g_{\xi_0}(\boldsymbol{\Psi}_{t-1})))$$

(7)

### 3.4.2 Posterior distribution over dictionaries

Similarly to the prior model, the variational posterior too is modeled as a Gaussian distribution parametrized by a neural network $f_\phi(\cdot) = [f_{\phi_1}^\mu \circ h_{\phi_0}(\cdot), f_{\phi_2}^{\sigma^2} \circ h_{\phi_0}(\cdot)]$ which outputs the mean and variance for the underlying probability distribution: $q_\phi(\boldsymbol{\Psi}_t | \hat{\boldsymbol{x}}_{t-1}, \boldsymbol{y}^i, \boldsymbol{\Phi}^i) = \mathcal{N}(\boldsymbol{\Psi}_t | \boldsymbol{\mu} = f_{\phi_1}^\mu(h_{\phi_0}(\hat{\boldsymbol{x}}_{t-1}, \boldsymbol{y}^i, \boldsymbol{\Phi}^i)); \boldsymbol{\sigma}^2 = f_{\phi_2}^{\sigma^2}(h_{\phi_0}(\hat{\boldsymbol{x}}_{t-1}, \boldsymbol{y}^i, \boldsymbol{\Phi}^i)))$. The posterior distribution is conditioned on the data, $\{\boldsymbol{y}^i, \boldsymbol{\Phi}^i\}$, as well as on the reconstructed signal at the previous layer, $\hat{\boldsymbol{x}}_{t-1}$. Therefore, the joint posterior probability over the dictionaries at each layer is given by:

$$q_\phi(\boldsymbol{\Psi}_{1:T} | \hat{\boldsymbol{x}}_{1:T}, \boldsymbol{y}^i, \boldsymbol{\Phi}^i) = \prod_{t=1}^{T} q_\phi(\boldsymbol{\Psi}_t | \hat{\boldsymbol{x}}_{t-1}, \boldsymbol{y}^i, \boldsymbol{\Phi}^i) \tag{8}$$

### 3.4.3 Likelihood model

At the heart o the reconstruction module there is the soft-thresholding block of A-DLISTA. Similarly to the prior and posterior, the likelihood distribution is modelled as a Gaussian parametrized by the output of a A-DLISTA block. Specifically, the likelihood network generates the mean vector only for the Gaussian distribution since we treat the standard deviation as a tunable hyper-parameter. Therefore, we interpret the reconstructed sparse vector at a given layer as the mean of the likelihood distribution. The joint log-likelihood distribution can then be formulated as:

$$\log p_\Theta(\hat{\boldsymbol{x}}_{1:T} | \boldsymbol{\Psi}_{1:T}, \boldsymbol{y}^i, \boldsymbol{\Phi}^i) = \sum_{t=1}^{T} \log \mathcal{N}(\boldsymbol{\mu}_t = \text{A-DLISTA}(\boldsymbol{\Psi}_{1:t}, \boldsymbol{y}^i, \boldsymbol{\Phi}^i; \Theta), \boldsymbol{\sigma}^2{}_t = \delta) \tag{9}$$

where $\delta$ is a hyper-parameter of the network.

We train all the different components of VLISTA, in an end-to-end fashion by the maximization of the Evidence Lower Bound (ELBO). The full objective function is given by:

$$\text{ELBO} = \sum_{t=1}^{T} \mathbb{E}_{\boldsymbol{\Psi}_{1:t} \sim q_\phi(\boldsymbol{\Psi}_{1:t} | \boldsymbol{y}^i, \boldsymbol{\Phi}^i, \hat{\boldsymbol{x}}_{0:t-1})} \Big[ \log p_\Theta(\boldsymbol{x}_t = \boldsymbol{x}_{gt}^i | \boldsymbol{\Psi}_{1:t}, \boldsymbol{y}^i, \boldsymbol{\Phi}^i) \Big] \tag{10}$$

$$- \sum_{t=2}^{T} \mathbb{E}_{\boldsymbol{\Psi}_{1:t-1} \sim q_\phi(\boldsymbol{\Psi}_{1:t-1} | \boldsymbol{y}^i, \boldsymbol{\Phi}^i, \hat{\boldsymbol{x}}_{t-1})} \Big[ D_{KL}\Big( q_\phi(\boldsymbol{\Psi}_t | \boldsymbol{y}^i, \boldsymbol{\Phi}^i, \hat{\boldsymbol{x}}_{t-1}) \, \| \, p_\xi(\boldsymbol{\Psi}_t | \boldsymbol{\Psi}_{t-1}) \Big) \Big]$$

$$- D_{KL}\Big( q_\phi(\boldsymbol{\Psi}_1 | \hat{\boldsymbol{x}}_0) \, \| \, p_\xi(\boldsymbol{\Psi}_1) \Big)$$

The first term in Equation 10 represents the likelihood contribution whilst the second and third terms account for the KL divergence. We report more details about models' architecture and the objective function in Appendix A and Appendix B, respectively.

## 4  Experimental Results

To assess the performance of the proposed approach, we employ three datasets, namely, MNIST, CIFAR10, and a synthetic one. We compare our models' performance against ISTA, LISTA Gregor & LeCun (2010), and BCS Ji et al. (2008). However, we do not consider other LISTA variations such as ALISTA Liu et al. (2019) or NALISTA Behrens et al. (2021) since assuming a varying measurement setup across the dataset requires solving an inner optimization problem to evaluate the $\mathbf{W}$ matrix for each data sample. As a result, training such models is extremely slow. Moreover, to prove the benefit of adaptivity, we conduct an ablation study on A-DLISTA by removing its augmentation network and making the parameters $\theta_t, \gamma_t$ learnable through backprop. We refer to the non-augmented version of A-DLISTA as DLISTA (see subsection 3.3 for more details). Hence, for DLISTA, $\theta_t$ and $\gamma_t$ cannot be adapted anymore to the specific input sensing matrix. For all models that we train, we consider

three layers. However, being ISTA a classical method with no learning properties, we also considered the results obtained using 1000 iterations. To consider a scenario with varying sensing matrices, we adopt the following procedure. For each data sample in the training and test sets, $x^i$, we generate a sensing matrix, $\Phi^i$, by randomly sampling its entries from a standard distribution. Subsequently, for each pair of sensing matrix and ground truth signal, we generate the corresponding observations as $y^i = \Phi^i \cdot x^i$. We report more details about the training of the models in Appendix B.

## 4.1 MNIST & CIFAR10

The first task we test our models against is image reconstruction considering the MNIST and CIFAR10 datasets. We report the results in terms of the Structural Similarity Index Measure (SSIM) considering the following setups. We fix the number of layers, or iterations, for all models to three and then we measured SSIM varying the number of measurements. To compute the observation vector ($y^i$) we generate a different sensing matrix, for each digit, by sampling its entries from a standard Gaussian distribution (more details about the data generation can be found in Appendix B). The results are reported in Table 1 and Table 2 for MNIST abd CIFAR10, respectively.

Table 1: MNIST SSIM (the higher the better) for different number of measurements. Top three rows concern non-Bayesian models whilst the bottom two report results for Bayesian approaches. For each of the two set of results, we highlight in bold the best performance.

| | SSIM ↑ | | | | |
|---|---|---|---|---|---|
| | number of measurements | | | | |
| | 1 | 10 | 100 | 300 | 500 |
| ISTA | 0.10 | 0.07 | 0.06 | 0.20 | 0.34 |
| ISTA (1000 iterations) | 0.10 | 0.07 | 0.12 | 0.46 | 0.76 |
| LISTA | 0.21 | 0.54 | 0.56 | 0.63 | 0.75 |
| DLISTA | 0.27 | 0.60 | 0.64 | 0.64 | 0.76 |
| A-DLISTA | **0.33** | **0.66** | **0.81** | **0.84** | **0.88** |
| BCS | 0.09 | 0.12 | 0.19 | 0.45 | 0.73 |
| VLISTA | **0.22** | **0.45** | **0.56** | **0.65** | **0.78** |

Table 2: CIFAR10 SSIM (the higher the better) for different number of measurements. Top three rows concern non-Bayesian models whilst the bottom two report results for Bayesian approaches. For each of the two set of results, we highlight in bold the best performance.

| | SSIM ↑ | | | | | | |
|---|---|---|---|---|---|---|---|
| | number of measurements | | | | | | |
| | 10 | 50 | 100 | 300 | 500 | 700 | 850 |
| ISTA | 0.001 | 0.004 | 0.009 | 0.025 | 0.035 | 0.040 | 0.041 |
| ISTA (1000 iterations) | 0.002 | 0.009 | 0.015 | 0.038 | 0.051 | 0.057 | 0.055 |
| LISTA | 0.008 | 0.227 | 0.271 | 0.597 | 0.658 | 0.716 | 0.779 |
| DLISTA | 0.348 | 0.458 | 0.472 | 0.581 | 0.647 | 0.713 | 0.778 |
| A-DLISTA | **0.581** | **0.684** | **0.717** | **0.776** | **0.799** | **0.812** | **0.852** |
| BCS | 0.005 | 0.028 | 0.046 | 0.114 | 0.194 | 0.286 | 0.359 |
| VLISTA | **0.265** | **0.286** | **0.464** | **0.617** | **0.692** | **0.731** | **0.774** |

From Table 1 and Table 2, we can draw the following conclusions. Concerning the non-Bayesian models, our A-DLISTA model outperforms all the others. Moreover, by comparing the performance of A-DLISTA with its non-augmented version, i.e., DLISTA, we see the benefits of using an augmentation network to make the model adaptive. Instead, concerning the Bayesian approaches, our VLISTA model outperforms BCS. Especially, we see that our models outperform others considering

a low number of measurements. Concerning the lower performance of VLISTA compared to A-DLISTA, we can mention a few reasons that could explain such behaviour. One contribution to such a difference might come from the noise that is naturally injected at training time due to the random sampling procedure to generate the dictionaries. Also, another contribution is represented by the amortization gap that affects all models based on amortized variational inference. However, although VLISTA shows lower performance than A-DLISTA, it is important to notice that it still performs better than BCS. Moreover, it can detect OODs, a characteristic that Bayesian models only possess.

## 4.2 Synthetic Dataset

To generate the synthetic dataset, we follow a similar protocol as inLiu & Chen (2019). First, we generate a sensing matrix and a sparsifying dictionary, for each data sample, by sampling their entries from a standard Gaussian distribution. Then, the components of the ground truth sparse signal, $z^*$, are sampled from a standard Gaussian distribution as well. Finally, some of the components of $z^*$ are set to zero as dictated by a Bernoulli distribution with $p = 0.1$. The overall dataset accounts for 1000 samples shared across the train and test sets. To compare the performance of different models, first, we draw the c.d.f of the Normalized Mean Square Error (NMSE) on the test set and then we compute the 40% quantile. Results are reported in Table 3. Similarly to the setup we used in the previous section, also for the synthetic dataset we fix the number of layers, or iterations, for each model to three and then we varied the number of measurements.

Table 3: NMSE's quantile (the lower the better) for different number of measurements. Top three rows concern non-Bayesian models whilst the bottom two report results for Bayesian approaches. For each of the two set of results, we highlight in bold the best performance.

| | Q=0.4 ↓ | | | | |
|---|---|---|---|---|---|
| | number of measurements | | | | |
| | 1 | 10 | 100 | 300 | 500 |
| ISTA | +0.38 | +0.05 | -0.13 | -1.90 | -3.28 |
| ISTA (1000 iterations) | +0.01 | -0.04 | -1.62 | -8.30 | -15.76 |
| LISTA | -0.01 | -1.60 | -1.28 | -2.82 | -4.37 |
| DLISTA | -3.12 | -5.22 | -7.18 | -10.62 | -17.92 |
| A-DLISTA | **-5.45** | **-14.82** | **-20.91** | **-19.47** | **-22.35** |
| BCS | +0.16 | +0.17 | +1.91 | +2.34 | +2.55 |
| VLISTA | **-3.67** | **-8.20** | **-13.94** | **-15.31** | **-14.02** |

By looking at Table 3 we can draw a similar conclusion as for the MNIST and CIFAR10 datasets.

## 4.3 Out Of Distribution detection

In this section, we focus on one of the most important differences among non-Bayesian models for solving inverse linear problems and VLISTA. Indeed, differently from any non-Bayesian approach to compressed sensing, VLISTA allows for quantifying uncertainties on the reconstructed signals which, in turn, enables OOD detection without the need to access ground truth data at inference time. Moreover, whilst other Bayesian approaches Ji et al. (2008); Zhou et al. (2014) usually focus on designing specific priors to satisfy the sparsity constraint on the reconstructed signal after marginalization, VLISTA completely overcomes such an issue as the thresholding operations is not affected by the the marginalization over dictionaries. To prove that VLISTA can detect OOD samples, we employ the MNIST dataset. First, we split the full dataset into two subsets named "Train", or In-Distribution (ID), and OOD. The ID subset contains images from three digits only, namely, 0, 3, and 7 (randomly chosen). Instead, the OOD subset contains images from all the other digits. Then, we split the ID partition into training and test sets and train VLISTA on the former one. Once trained, we evaluate the model performance by considering reconstructions from the test set (ID) and the OOD partition. We reconstruct 100 times every single image, sampling every time a new dictionary. Subsequently, as a summarizing statistics we compute the variance's c.d.f. of the per-pixel standard

deviation ($var_{\sigma_{pp}}$) across reconstructions. Subsequently, to assess whether a given digit belongs to the ID or OOD distribution, we compute the p-value for $var_{\sigma_{pp}}$ by employing the two-sample t-test.

Moreover, to assess whether the OOD detection is robust to measurement noise, we repeat the same test for different levels of noise. As a baseline for the current task, we consider BCS. Due to the different nature of the BCS framework, we employ a slightly different procedure to evaluate the p-values for it. Specifically, we use the same ID and OOD splits as for VLISTA. However, for BCS, we consider the c.d.f. of the reconstruction error that is evaluated by the model itself. The rest of the procedure is the same as for VLISTA. We report the results for OOD detection in Figure 2. As we can see from the figure, VLISTA outperforms BCS for each level of noise showing a lower p-value than BCS which corresponds to a higher rejection power. As expected, by increasing the level of noise we observe a larger p-value meaning that the OOD rejection becomes harder for more noisy data. However, we can see that whilst VLISTA is still

Figure 2: p-value for OOD rejection as a function of the noise level. The green line represents a reference p-value equal to 0.05.

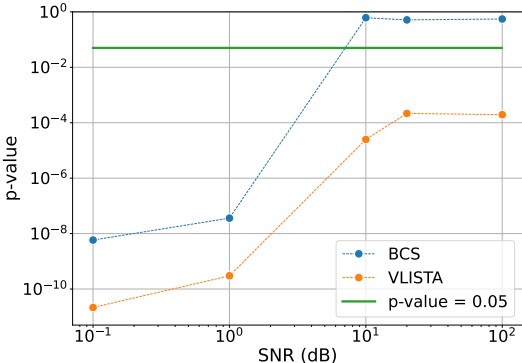

capable of detecting OOD samples, BCS fails in doing so when the Signal-to-Noise Ratio (SNR), expressed in decibels, is greater than 10. As a reference point to define whether the model is correctly rejecting OOD samples or not, we report in Figure 2 the 5% line for the p-value. Such a value is typically used as a reference in hypothesis testing to decide whether or not to reject the null hypothesis.

## 5    Conclusion

We report about a variational approach, dubbed VLISTA, to solve the dictionary learning and the sparse recovery problems jointly. Typically, compressed sensing frameworks assume the existence of a ground truth dictionary used to reconstruct the signal. Furthermore, in state-of-the-art LISTA-like models, a stationary measurement setup is usually considered. In our work, we relax both assumptions. First, we show that it is possible to design a soft-thresholding algorithm, termed A-DLISTA, that can handle different sensing matrices and that can adapt its parameters to the given data instance. We theoretically justify the use of an augmentation network which adapts the threshold and step size for each layer based on the current input and the learned dictionary. Finally, we also relax the hypothesis concerning the existence of a ground truth dictionary by introducing a probability distribution for it. Given such an assumption, we formulate the VLISTA variational framework to solve the compressed sensing task. We report results for both our models, A-DLISTA and VLISTA, concerning non-Bayesian and Bayesian approaches to solve the sparse recovery and dictionary learning problems jointly. We empirically show that the adaptation capability of A-DLISTA results in a boost in performance compared to ISTA and LISTA models, in a non-static measurements scenario. Although we observe that in terms of reconstruction, VLISTA does not outperform A-DLISTA, the variational framework enables us to evaluate uncertainties over the reconstructed signals useful to detect OOD. On the other hand, none of the LISTA-like models allows for such a task. Moreover, differently from other Bayesian approaches to compressed sensing, VLISTA does not need to design specific priors to retain sparsity after marginalization of the reconstructed sparse signal; the averaging operation concerns the sparsifying dictionary instead of the sparse signal itself.

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

# A   Architecture Details

In this section we report the details of the architecture for our proposed models.

Concerning A-DLISTA, as we show in Figure 1, the reconstruction network, i.e., blue blocks, is an unfolded ISTA-like model with parametrized dictionary $\Psi_t$. Each layer is characterized by its own dictionary which used to both reconstruct the sparse vector and as an input for the augmentation network. As mentioned in subsection 3.3, the augmentation model, red block in Figure 1, takes as input the measurement matrix, $\Phi^i$, and the dictionary at a given reconstruction layer $t$, $\Psi_t$, and generates the adaptive parameters $\{\gamma_t, \theta_t\}$ for the $t$-th layer. We show the architecture for the augmentation network in Figure 3.

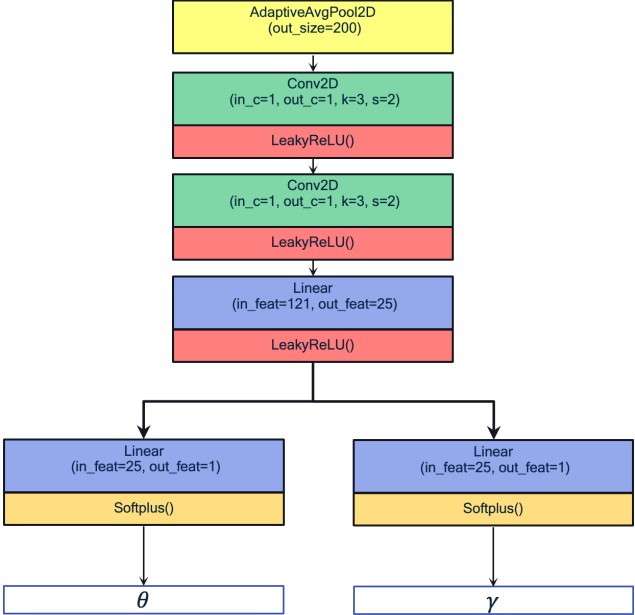

Figure 3: Augmentation network for A-DLISTA. Each red block in Figure 1 corresponds to the shown model.

Concerning VLISTA, as introduced in subsection 3.4, it comprises three different models: prior, posterior, and likelihood. Concerning the likelihood model, it is assumed to represent a gaussian distribution whose mean is parametrized by means of the A-DLISTA model (subsubsection 3.4.3). Instead, the prior (subsubsection 3.4.1) and posterior (subsubsection 3.4.2) models are implemented using an encoder-decoder scheme based on convolutional layers. We report in Figure 4 the architecture for the prior and posterior mdoels.

Finally, we report in Figure 5 the graphical model for the posterior and conditional prior in the left and right plot, respectively.

# B   Implementation and Training Details

We report in this section a few details about the implementation and training of the A-DLISTA and VLISTA models. We implemented both using the Lightning framework. As we mentioned in the main body of the manuscript, ISTA and LISTA require a known dictionary in order to reconstruct the non-sparse signal. Concerning the three datasets that we consider, we define the dictionaries in the following way: canonical for MNIST (since MNIST is already sparse) with 784 atoms and wavelet for CIFAR10 with 1024 atoms. Regarding the synthetic dataset, we randomly generated the dictionary from a standard distribution and we consider 765 atoms. Concerning A-DLISTA, we trained both the reconstruction and the augmentation network, blue and red blocks in Figure 1, respectively, end-to-end using the Adam optimizer. We set the initial learning rate to $1.e^{-2}$ and $1.e^{-3}$ for the reconstruction and augmentation network respectively, and we dropped its value by a factor 10

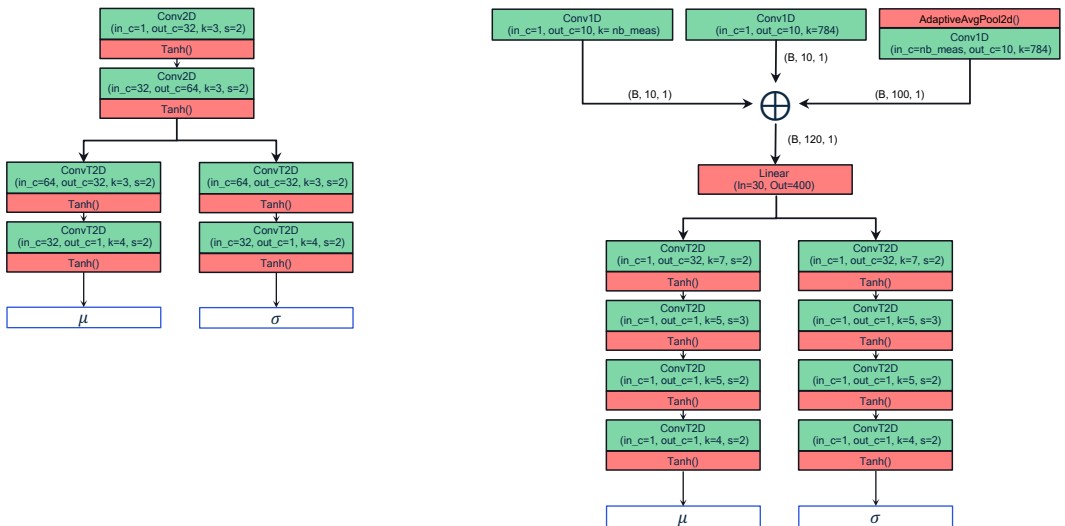

Figure 4: Left: prior network architecture. Right: posterior network architecture. For the posterior model we show in the figure the output shape from each of the three head. Such a structure is necessary since the posterior model accepts as input three quantities, namely, the observations, the sensing matrix, and the reconstruction from the previous layer which are characterized by different shapes. The term "B" indicates the batch size.

Figure 5: Graphical model of the Variational LISTA model - dependencies on $\boldsymbol{y}^i$, $\boldsymbol{\Phi}^i$ are factored out for simplicity. The sampling is done only based on the posterior $q_\phi(\boldsymbol{\Psi}_t|\hat{\boldsymbol{x}}_{t-1}, \boldsymbol{y}^i, \boldsymbol{\Phi}^i)$. The dashed line shows variational approximations

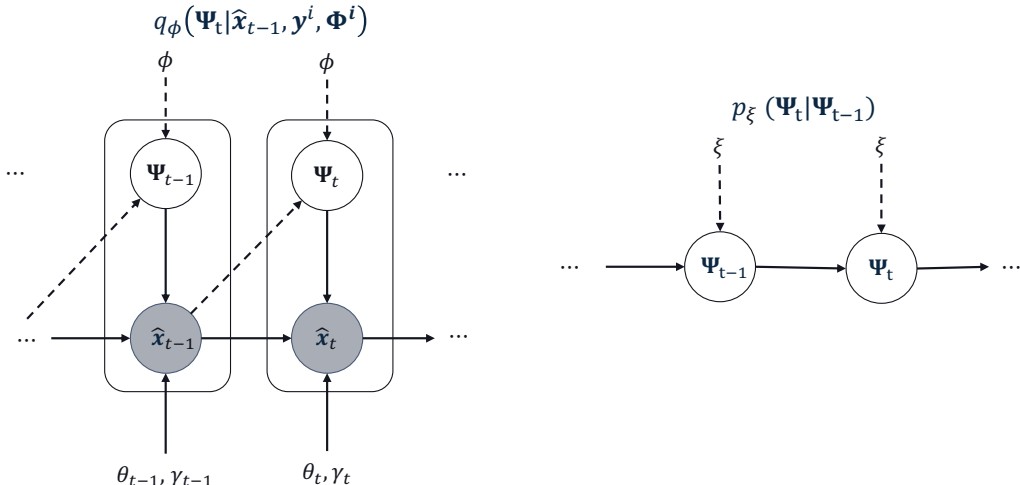

every time the loss stopped to improve for more than 30 training steps. Moreover, we set the weight decay to $5.e^{-4}$ and the batch size to 128. We applied the same scheme across all the dataets we used. As the objective function, we used the MSE between the ground truth signal and the reconstructed one for image datasets and the NMSE for the synthetic dataset, respectively.

Concerning the VLISTA training, similarly to what we did for A-DLISTA, we train the full model end-to-end. However, compared to A-DLISTA, in this case the hyperparameter space has a much higher dimension. Therefore, we employed the Tune library for hyperparameter search. Specifically, we used HyperOptSearch and ASHAScheduler as searcher and scheduler, respectively. We set the learning rates to $7.e^{-3}$, $5.e^{-3}$, and $1.e^{-4}$ for the likelihood, posterior, and prior models, respectively. Also in this case, we use a scheduler for reducing the learning rate similarly to what we did for A-DLISTA. Concerning the objective function, we maximize the ELBO as typically done for such a

type of models. Moreover, we set the weight for the KL divergence to $1.e^{-3}$. We report in Equation 11 details about the obejctive.

$$\log\Big(p(\boldsymbol{x}_{1:T} = \boldsymbol{x}_{gt}^i|\boldsymbol{y}^i, \boldsymbol{\Phi}^i)\Big) = \log \int p(\boldsymbol{x}_{1:T} = \boldsymbol{x}_{gt}^i|\boldsymbol{\Psi}_{1:T}, \boldsymbol{y}^i, \boldsymbol{\Phi}^i)p(\boldsymbol{\Psi}_{1:T})d\boldsymbol{\Psi}_{1:T} \tag{11}$$

$$= \log \int \frac{p(\boldsymbol{x}_{1:T} = \boldsymbol{x}_{gt}^i|\boldsymbol{\Psi}_{1:T}, \boldsymbol{y}^i, \boldsymbol{\Phi}^i)p(\boldsymbol{\Psi}_{1:T})q(\boldsymbol{\Psi}_{1:T}|\boldsymbol{y}^i, \boldsymbol{\Phi}^i, \hat{\boldsymbol{x}}_{1:T})}{q(\boldsymbol{\Psi}_{1:T}|\boldsymbol{y}^i, \boldsymbol{\Phi}^i, \hat{\boldsymbol{x}}_{1:T})}d\boldsymbol{\Psi}_{1:T}$$

$$\geq \int q(\boldsymbol{\Psi}_{1:T}|\boldsymbol{y}^i, \boldsymbol{\Phi}^i, \hat{\boldsymbol{x}}_{1:T}) \log\frac{p(\boldsymbol{x}_{1:T} = \boldsymbol{x}_{gt}^i|\boldsymbol{\Psi}_{1:T}, \boldsymbol{y}^i, \boldsymbol{\Phi}^i)p(\boldsymbol{\Psi}_{1:T})}{q(\boldsymbol{\Psi}_{1:T}|\boldsymbol{y}^i, \boldsymbol{\Phi}^i, \hat{\boldsymbol{x}}_{1:T})}d\boldsymbol{\Psi}_{1:T}$$

$$= \int q(\boldsymbol{\Psi}_{1:T}|\boldsymbol{y}^i, \boldsymbol{\Phi}^i, \hat{\boldsymbol{x}}_{1:T}) \log p(\boldsymbol{x}_{1:T} = \boldsymbol{x}_{gt}^i|\boldsymbol{\Psi}_{1:T}, \boldsymbol{y}^i, \boldsymbol{\Phi}^i)d\boldsymbol{\Psi}_{1:T}$$

$$+ \int q(\boldsymbol{\Psi}_{1:T}|\boldsymbol{y}^i, \boldsymbol{\Phi}^i, \hat{\boldsymbol{x}}_{1:T}) \log \frac{p(\boldsymbol{\Psi}_{1:T})}{q(\boldsymbol{\Psi}_{1:T}|\boldsymbol{y}^i, \boldsymbol{\Phi}^i, \hat{\boldsymbol{x}}_{1:T})}d\boldsymbol{\Psi}_{1:T}$$

$$= \sum_{t=1}^{T}\mathbb{E}_{\boldsymbol{\Psi}_{1:t}\sim q(\boldsymbol{\Psi}_{1:t}|\boldsymbol{y}^i, \boldsymbol{\Phi}^i, \hat{\boldsymbol{x}}_{0:t-1})}\Big[\log p(\boldsymbol{x}_t = \boldsymbol{x}_{gt}^i|\boldsymbol{\Psi}_{1:t}, \boldsymbol{y}^i, \boldsymbol{\Phi}^i)\Big]$$

$$- \sum_{t=2}^{T}\mathbb{E}_{\boldsymbol{\Psi}_{1:t-1}\sim q(\boldsymbol{\Psi}_{1:t-1}|\boldsymbol{y}^i, \boldsymbol{\Phi}^i, \hat{\boldsymbol{x}}_{t-1})}\Big[D_{KL}\Big(q(\boldsymbol{\Psi}_t|\boldsymbol{y}^i, \boldsymbol{\Phi}^i, \hat{\boldsymbol{x}}_{t-1}) \parallel p(\boldsymbol{\Psi}_t|\boldsymbol{\Psi}_{t-1})\Big)\Big]$$

$$- D_{KL}\Big(q(\boldsymbol{\Psi}_1|\hat{\boldsymbol{x}}_0) \parallel p(\boldsymbol{\Psi}_1)\Big)$$

Finally, to give an overall overview of the training scheme for VLISTA, Figure 6 shows a diagram of the full training pipeline.

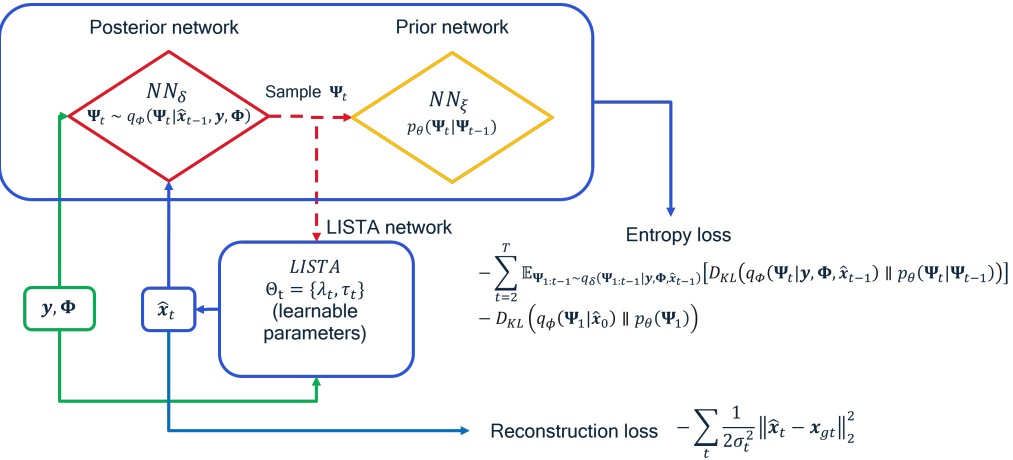

Figure 6: Variational LISTA oveall architecture and training pipeline. Note that to simplify the figure, we did not report the index for the data sample ($i$) but only for the iteration ($t$).

## C   Derivation for Theorem 3.1

Convergence proofs of ISTA type models involve two steps in general. First, it is investigated how the support is found and locked in, and second how the error shrinks at each step. We focus on these two steps, which matter mainly for our architecture design. Our analysis is similar in nature to Chen et al. (2018); Aberdam et al. (2021), however it differs from Aberdam et al. (2021) in considering unknown dictionaries and from Chen et al. (2018) in both considered architecture and varying sensing matrix. In what follows, we consider noiseless setting. However, the results can be extended to noisy setups by adding additional terms containing noise norm similar to Chen et al. (2018). We make following assumptions:

1. There is a ground-truth (unknown) dictionary $\boldsymbol{\Psi}_o$ such that $\boldsymbol{x}_* = \boldsymbol{\Psi}_o \boldsymbol{z}_*$.
2. As a consequence, $\boldsymbol{y} = \boldsymbol{\Phi}\boldsymbol{\Psi}_o \boldsymbol{z}_*$.
3. We assume that $\boldsymbol{z}_*$ is sparse with its support contained in $S$. In other words: $z_{i,*} = 0$ for $i \in S^c$.

As a first step, we fix the sensing matrix $\boldsymbol{\Phi}$ and conduct the analysis. First define the following:

$$\tilde{\mu} := \max_{1 \le i \ne j \le N} \left| ((\boldsymbol{\Phi}\boldsymbol{\Psi}_t)_i)^\top (\boldsymbol{\Phi}\boldsymbol{\Psi}_t)_j \right| \tag{12}$$

$$\tilde{\mu}_2 := \max_{1 \le i,j \le N} \left| ((\boldsymbol{\Phi}\boldsymbol{\Psi}_t)_i)^\top (\boldsymbol{\Phi}(\boldsymbol{\Psi}_t - \boldsymbol{\Psi}_o))_j \right| \tag{13}$$

$$\delta(\gamma) := \max_i \left| 1 - \gamma \left\| (\boldsymbol{\Phi}\boldsymbol{\Psi}_t)_i \right\|_2^2 \right| \tag{14}$$

The main step of soft-thresholding algorithm is given as follows:

$$\boldsymbol{z}_t = \eta_{\theta_t} \left( \boldsymbol{z}_{t-1} + \gamma_t (\boldsymbol{\Phi}\boldsymbol{\Psi}_t)^\top (\boldsymbol{y} - \boldsymbol{\Phi}\boldsymbol{\Psi}_t \boldsymbol{z}_{t-1}) \right), \tag{15}$$

with entry-wise relation given by

$$z_{t,i} = \eta_{\theta_t} \left( z_{t-1,i} + \gamma_t ((\boldsymbol{\Phi}\boldsymbol{\Psi}_t)_i)^\top (\boldsymbol{y} - \boldsymbol{\Phi}\boldsymbol{\Psi}_t \boldsymbol{z}_{t-1}) \right). \tag{16}$$

Using the assumptions we have:

$$\begin{aligned} (\boldsymbol{\Phi}\boldsymbol{\Psi}_t)^\top (\boldsymbol{y} - \boldsymbol{\Phi}\boldsymbol{\Psi}_t \boldsymbol{z}_{t-1}) &= (\boldsymbol{\Phi}\boldsymbol{\Psi}_t)^\top (\boldsymbol{\Phi}\boldsymbol{\Psi}_o \boldsymbol{z}_* - \boldsymbol{\Phi}\boldsymbol{\Psi}_t \boldsymbol{z}_{t-1}) \\ &= (\boldsymbol{\Phi}\boldsymbol{\Psi}_t)^\top \boldsymbol{\Phi}(\boldsymbol{\Psi}_o \boldsymbol{z}_* - \boldsymbol{\Psi}_t \boldsymbol{z}_{t-1}). \end{aligned} \tag{17}$$

### C.1   Locking the support

First, we show under what conditions, the algorithm locks on the support. Suppose that the support of $\boldsymbol{z}_{t-1}$ is already the same as $\boldsymbol{z}_*$, namely $\text{supp}(\boldsymbol{z}_{t-1}) = \text{supp}(\boldsymbol{z}_*) = S$. Consider $i \in S^c$. We have

$$z_{t,i} = \eta_{\theta_t} \left( \gamma_t ((\boldsymbol{\Phi}\boldsymbol{\Psi}_t)_i)^\top (\boldsymbol{y} - \boldsymbol{\Phi}\boldsymbol{\Psi}_t \boldsymbol{z}_{t-1}) \right). \tag{18}$$

To lock the support, we need to guarantee that:

$$\left| \gamma_t ((\boldsymbol{\Phi}\boldsymbol{\Psi}_t)_i)^\top (\boldsymbol{y} - \boldsymbol{\Phi}\boldsymbol{\Psi}_t \boldsymbol{z}_{t-1}) \right| \le \theta_t. \tag{19}$$

We have:

$$\begin{aligned} ((\boldsymbol{\Phi}\boldsymbol{\Psi}_t)_i)^\top \boldsymbol{\Phi}(\boldsymbol{\Psi}_o \boldsymbol{z}_* - \boldsymbol{\Psi}_t \boldsymbol{z}_{t-1}) =& ((\boldsymbol{\Phi}\boldsymbol{\Psi}_t)_i)^\top \boldsymbol{\Phi}(\boldsymbol{\Psi}_t \boldsymbol{z}_* - \boldsymbol{\Psi}_t \boldsymbol{z}_{t-1}) \\ &+ ((\boldsymbol{\Phi}\boldsymbol{\Psi}_t)_i)^\top \boldsymbol{\Phi}(\boldsymbol{\Psi}_o \boldsymbol{z}_* - \boldsymbol{\Psi}_t \boldsymbol{z}_*) \end{aligned} \tag{20}$$

$$= \sum_{j \in S} ((\boldsymbol{\Phi}\boldsymbol{\Psi}_t)_i)^\top (\boldsymbol{\Phi}\boldsymbol{\Psi}_t)_j (z_{*,j} - z_{t-1,j}) \tag{21}$$

$$+ ((\boldsymbol{\Phi}\boldsymbol{\Psi}_t)_i)^\top \boldsymbol{\Phi}(\boldsymbol{\Psi}_o \boldsymbol{z}_* - \boldsymbol{\Psi}_t \boldsymbol{z}_*) \tag{22}$$

We can bound the norm by:

$$\left| \sum_{j \in S} ((\boldsymbol{\Phi}\boldsymbol{\Psi}_t)_i)^\top (\boldsymbol{\Phi}\boldsymbol{\Psi}_t)_j (z_{*,j} - z_{t-1,j}) \right| \le \sum_{j \in S} \left| ((\boldsymbol{\Phi}\boldsymbol{\Psi}_t)_i)^\top (\boldsymbol{\Phi}\boldsymbol{\Psi}_t)_j \right| \left| (z_{*,j} - z_{t-1,j}) \right| \tag{23}$$

$$\le \tilde{\mu} \left\| \boldsymbol{z}_* - \boldsymbol{z}_{t-1} \right\|_1, \tag{24}$$

where we use the definition of mutual coherence for the upper bound.

The other norm is bounded by

$$\left|((\mathbf{\Phi\Psi}_t)_i)^\top \mathbf{\Phi}(\mathbf{\Psi}_o \boldsymbol{z}_* - \mathbf{\Psi}_t \boldsymbol{z}_*)\right| = \left|\sum_{j \in S}((\mathbf{\Phi\Psi}_t)_i)^\top (\mathbf{\Phi}(\mathbf{\Psi}_o - \mathbf{\Psi}_t))_j z_{j,*}\right| \tag{25}$$

$$\leq \sum_{j \in S}\left|((\mathbf{\Phi\Psi}_t)_i)^\top (\mathbf{\Phi}(\mathbf{\Psi}_o - \mathbf{\Psi}_t))_j\right| |z_{j,*}| \tag{26}$$

$$\leq \tilde{\mu}_2 \left\|\boldsymbol{z}_*\right\|_1 . \tag{27}$$

Therefore, we obtain the following sufficient condition for locking the support:

$$\gamma_t \left(\tilde{\mu} \left\|\boldsymbol{z}_* - \boldsymbol{z}_{t-1}\right\|_1 + \tilde{\mu}_2 \left\|\boldsymbol{z}_*\right\|_1\right) \leq \theta_t \tag{28}$$

If this condition is satisfied, we get to lock the support.

## C.2 Controlling the errors

For $i \in S$, we have:

$$|z_{t,i} - z_{*,i}| \leq \left|z_{t-1,i} + \gamma_t((\mathbf{\Phi\Psi}_t)_i)^\top (\boldsymbol{y} - \mathbf{\Phi\Psi}_t \boldsymbol{z}_{t-1}) - z_{*,i}\right| + \theta_t. \tag{29}$$

We start again with equation 16 but with $i \in S$:

$$z_{t-1,i} + \gamma_t((\mathbf{\Phi\Psi}_t)_i)^\top (\boldsymbol{y} - \mathbf{\Phi\Psi}_t \boldsymbol{z}_{t-1}) =$$
$$z_{t-1,i} + \gamma_t(\sum_{j \in S}((\mathbf{\Phi\Psi}_t)_i)^\top (\mathbf{\Phi\Psi}_t)_j (\boldsymbol{z}_{*,j} - \boldsymbol{z}_{t-1,j}) + ((\mathbf{\Phi\Psi}_t)_i)^\top \mathbf{\Phi}(\mathbf{\Psi}_o \boldsymbol{z}_* - \mathbf{\Psi}_t \boldsymbol{z}_*))$$

For the first part, we get:

$$z_{t-1,i} + \gamma_t \sum_{j \in S}((\mathbf{\Phi\Psi}_t)_i)^\top (\mathbf{\Phi\Psi}_t)_j (z_{*,j} - z_{t-1,j}) =$$
$$(\boldsymbol{I} - \gamma_t(\mathbf{\Phi\Psi}_t)_i)^\top (\mathbf{\Phi\Psi}_t)_i))z_{t-1,i} + \gamma_t(\mathbf{\Phi\Psi}_t)_i)^\top (\mathbf{\Phi\Psi}_t)_i)z_{*,i} + \gamma_t \sum_{j \in S, j \neq i}((\mathbf{\Phi\Psi}_t)_i)^\top (\mathbf{\Phi\Psi}_t)_j (z_{*,j} - z_{t-1,j}).$$

Therefore:

$$\left|z_{t-1,i} + \gamma_t((\mathbf{\Phi\Psi}_t)_i)^\top (\boldsymbol{y} - \mathbf{\Phi\Psi}_t \boldsymbol{z}_{t-1}) - z_{*,i}\right| \leq \left|(1 - \gamma_t(\mathbf{\Phi\Psi}_t)_i)^\top (\mathbf{\Phi\Psi}_t)_i))(z_{t-1,i} - z_{*,i})\right|$$
$$+ \gamma_t \left|\sum_{j \in S, j \neq i}((\mathbf{\Phi\Psi}_t)_i)^\top (\mathbf{\Phi\Psi}_t)_j (z_{*,j} - z_{t-1,j})\right| + \gamma_t \left|((\mathbf{\Phi\Psi}_t)_i)^\top \mathbf{\Phi}(\mathbf{\Psi}_o \boldsymbol{z}_* - \mathbf{\Psi}_t \boldsymbol{z}_*))\right|$$
$$\leq \delta(\gamma_t) \left|(z_{t-1,i} - z_{*,i})\right| + \gamma_t \sum_{j \in S, j \neq i} \tilde{\mu} |z_{*,j} - z_{t-1,j}| + \gamma_t \tilde{\mu}_2 \left\|\boldsymbol{z}_*\right\|_1$$

Therefore, we have:

$$\left\|\boldsymbol{z}_{S,t} - \boldsymbol{z}_*\right\|_1 = \sum_{i \in S} |z_{t,i} - z_{*,i}| \leq$$
$$\leq (\delta(\gamma_t) + \gamma_t \tilde{\mu}(|S| - 1)) \left\|\boldsymbol{z}_{S,t-1} - \boldsymbol{z}_*\right\|_1 + \gamma_t \tilde{\mu}_2 |S| \left\|\boldsymbol{z}_*\right\|_1 + |S|\theta_t.$$

Note that this analysis shows that we strive for having smaller $\tilde{\mu}_2$, namely being close to the dictionary, small enough $\theta_t$ as iterations grow, and small $\tilde{\mu}$ and $\delta(\gamma_t)$.

In general, if we have

$$\gamma_t \left(\tilde{\mu} \left\|\boldsymbol{z}_* - \boldsymbol{z}_{t-1}\right\|_1 + \tilde{\mu}_2 \left\|\boldsymbol{z}_*\right\|_1\right) \leq \theta_t. \tag{30}$$

Then we get the final result:

$$\left\|\boldsymbol{z}_t - \boldsymbol{z}_*\right\|_1 = \sum_{i \in S} |z_{t,i} - z_{*,i}| \leq (\delta(\gamma_t) + \gamma_t \tilde{\mu}(|S| - 1)) \left\|\boldsymbol{z}_{t-1} - \boldsymbol{z}_*\right\|_1 + \gamma_t \tilde{\mu}_2 |S| \left\|\boldsymbol{z}_*\right\|_1 + |S|\theta_t.$$

