# OpenReview forum: "Variational Learning ISTA"
_ICLR.cc/2023/Conference — Submitted to ICLR 2023_

### Official Review · Reviewer_uswp · 2022-10-23

**Confidence:** 3
**Correctness:** 4
**Technical Novelty And Significance:** 3
**Empirical Novelty And Significance:** 2
**Recommendation:** 6

**Clarity, Quality, Novelty And Reproducibility:**

Some specific points in the paper are not clear:

I do not see how LISTA models assume that the learned dictionary is known a priori - my understanding is that this is part of what LISTA learns. It seems at times that the authors are combining the concept of "fixed dictionary" with "known dictionary", e.g., the last sentence of Section 3.2. Perhaps the authors mean to say that the sparse representation of the signal obtained by LISTA is not useful on its own without knowing the dictionary that can translate it into the actual signal?

In page 3, a matrix A is described such that $A^TA = \Phi\Psi$, but I do not believe that $\Phi\Psi$ would be a square matrix when compressed sensing is applied.

The performance characterization metrics in (4-6) appear to depend on the iteration count $t$, but the dependence is not referenced in the notation.

In eq. (5), it appears $\Psi$ should be $\Psi_t$ or $\Psi_o$ in the first part of the right hand side of the equation, given that no single dictionary is assumed?

In Theorem 3.1, the assumption includes "$\mathrm{supp}(z^*)=\mathrm{supp}(z_{t-1})$" and the conclusion says "$\mathrm{supp}(z_{t-1}) \subseteq \mathrm{supp}(z^*)$", which appears obvious to me. Is there a typo?

In Fig. 1, the dictionary $\Psi^t$ appears as an output of each blue block, and it is not clear how that is the case. There is also little detail on the augmentation network (red blocks) that produces the parameters. Note also the discrepancy in notation $\Psi_t$ vs. $\Psi^t$.

Some of the notation in Section 3.4 describes distribution for matrices and vectors in terms of scalar PDFs - if this is an abuse of notation, its meaning should be stated explicitly.

The training of the augmentation networks that need to take into account the sparsity matrices and dictionaries seems to be a significant undertaking. It is not clear how these networks are trained for the experiments.

**Strength And Weaknesses:**

The paper considers a novel setting that is more flexible than the common assumption in the literature.

I am personally not convinced regarding the impact of an approach that tolerates variable sensing matrices. When compressed sensing architectures exhibit different sensing operators for different data, it is usually because there is little control over the behavior of the operator, so it would be difficult to accurately obtain the operators involved. Perhaps the authors can elaborate in Section 3.2 regarding the problems where the matrix $\Phi$ can change for each sample but is known in each case.

The approach that includes variational dictionary learning, described in Sections 3.4.2-3 apparently assumes that the sensing matrix $\Phi$ is known, which counters the setup where LISTA is applied (in addition to the point above) and the matrices involved in LISTA are "learned". It is not clear how one would infer both $\Phi$ and the dictionary $\Psi$ simultaneously. The authors should also clarify whether the sensing matrix needs to be known as it varies in each case.

The experimental results are limited - for such a flexible prior more challenging datasets should be attempted. Other papers that use data-driven priors in compressed sensing, such as GANs, do so. Furthermore, the only baseline used that employs a data-driven model is LISTA, a method that has driven significant research into improved variations; thus, the comparison against older algorithms BCS (2008) and ISTA (2010) is not informative. Finally, if the authors take into account the computation time for ALISTA and NALISTA in evaluating the matrix $W$ for each sample to discount them from the comparison, they should compare this against the training time for the deep learning networks used in this and competing papers.


**Summary Of The Paper:**

The paper proposes a version of LISTA where the update parameters can be made specific to each data sample even when the sensing matrix is not the same among samples. The paper then also proposes a variational method to dictionary learning to be integrated with the former approach.


**Summary Of The Review:**

I have several questions regarding clarity on the details of the algorithm and the significance of the experimental results that prevent me from recommending acceptance. Given the wealth of approaches based on data-driven models and compressed sensing, the numerical results need to be more compelling.

Post-rebuttal: The authors have addressed the majority of my questions satisfactorily

---

> ### Author Response · Authors · 2022-11-18
> **Thank you for your detailed review**
>
> In what follows we try to address all the concerns raised by the reviewer
>
> Q: I am personally not...
>
> A: We can report about two relevant examples regarding a varying, but known, sensing matrix. The reconstruction of MRI images for which the acquisition step accounts for an adaptive process. Here, the sensing matrix is sampled from a known distribution. Then, given the known (sampled) sensing matrix, the signal is reconstructed. However, given that the process is adaptive, each data sample is characterized by a varying, but known, sensing matrix [1, 2].
> As a second example, we can report on MIMO systems. In such a case, the receiver, e.g., a mobile device, and the transmitter, e.g., a base station, are characterized by the so-called beamforming codebooks from which the sensing matrix is obtained as their Kronecker product. However, in reality, those matrices might change across base stations, yielding a different but known sensing matrix [3].
> [1]Y. T. et al., 2021. End-to-End Sequential Sampling and Reconstruction for MRI.
> [2]B. T. et al., 2020. Experimental design for MRI by greedy policy search.
> [3]R.-F. J. et al., 2018. Frequency-domain compressive channel estimation for frequency-selective hybrid millimeter wave MIMO systems.
>
> Q: The approach that...
>
> A: In sections 3.4.2-3, we define the posterior over dictionaries and the likelihood for the reconstruction task, respectively. In equations (8) and (9), the sensing matrix, $\Phi$, is used to condition the posterior and likelihood. Therefore,$\Phi$ is not inferred. It is used as part of the input to the posterior and likelihood models. In LISTA, $\Phi$ is not learned. LISTA learns a parametrization of the relation between the sparse representation of the gt signal and its observations. Therefore, when such a relations changes from one sample to another, since $\Phi$ varies, LISTA must interpolate across all the different relations between signal and observations. Thus, obtaining non-optimal performance conditioned on a given sensing matrix. Moreover, to reconstruct the ground truth signal, LISTA needs to know the dictionary since the model is designed to reconstruct the sparse representation only.
>
> Q: The experimental results...
>
> A: We added two new baselines to our results and tested our models on CIFAR10 too. As from the paper, we did not consider ALISTA nor NALISTA due to the difficulties in training those models with a varying sensing matrix. These models require first to solve an optimization problem to find the analytic weight matrix that, in our case, must be specific to each sensing matrix. We conducted preliminary tests and we did not observe any advantage compared to LISTA. However, in case the paper will be accepted and the reviewer will be interested in the results, we could add them for camera ready.
>
> Q:  I do not see how...
>
> A: We try to disentangle the two concepts. That is why we propose two approaches, A-DLISTA and VLISTA. While the former learns the dictionary, meaning that it assumes that the dictionary is unknown but fixed, the latter learns a distribution over dictionaries. Thus, without assuming that the dictionary is known nor fixed. Concerning LISTA, it assumes the existence of a dictionary. Indeed, this is also explicitly stated in the original LISTA paper in Sec 1.1 and 4.
>
> Q: In page...
>
> A: That is a typo. We corrected the mistake in the updated version of the manuscript.
>
> Q: The performance...
>
> A: To be precise, $\tilde{\mu}$, $\tilde{\mu}_2$ and $\delta(\gamma)$ are all functions of $\Phi$ and $\Psi_t$. However, since we focused on a single layer and a fixed sensing matrix, we dropped this dependence to simplify the notation. We clarified that in the paper.
>
> Q: In eq. (5)...
>
> A: The correct choice is $\Psi_t$, which is included in the main paper.
>
> Q: In Theorem 3.1...
>
> A:The correct expression is $supp(\bf z_t)$ $ \subseteq$ $supp(\bf z_*)$. We have modified it in the main paper.
>
> Q: In Fig. 1...
>
> A: The dictionary is not the output of s blue block. It represents the dictionary that is learnt at that specific layer. We clarify it in the caption of the image. Moreover, we fixed the notation by using lower indices to indicate the iteration index while the upper index refers to the data sample index. The augmentation network is trained end-to-end with the reconstruction model. We added more details about the training and the architecture of the augmentation architecture in the Appendix B of the revised manuscript.
>
> Q: Some of the notation...
>
> A: Indeed, there is an abuse of notation. We have revised our manuscript appropriately.
>
> Q: The training of...
>
> A: We added more details about the training procedure in Appendix B of the revised version of the manuscript.
>
> Q: I have several...
>
> A: We addressed most of the concerns by adding more details about training procedures, model architecture, and data generation. Moreover, we expanded our results by introducing two new baselines and testing our models on a new dataset.

---

> ### Author Response · Authors · 2022-12-12
> **Addressing concerns**
>
> Dear review uswp,
>
> As the discussion period is closing, we would like to thank you again for the time and effort you dedicated to reviewing our work. We tried to address your concerns at our best and updated the manuscript accordingly. We are looking forward to your comments on our rebuttal.
>
> **Fixed Dictionary and Known Dictionary:**
> In our work, we try to disentangle the two concepts. That is why we propose two approaches, A-DLISTA and VLISTA. While the former learns the dictionary, meaning that it assumes that the dictionary is unknown but fixed, the latter learns a distribution over dictionaries. Thus, without assuming that the dictionary is known nor fixed.
>
> **Numerical Results:**
> Following the reviewers' suggestions, we conducted experiments on an additional dataset and added new comparisons with baselines. All the new results confirmed the conclusions we had previously drawn. Our A-DLISTA outperforms all the non-Bayeisan models while VLISTA reached better performance than BCS.
>
> **Augmentation Network:**
> To make our results clearer to the readers and reproducible from other researchers, we expanded Appendices A and B by reporting several details about the architecture and training procedure of the augmentation as well as the prior and posterior models.

---

> > ### Comment · Reviewer_uswp · 2022-12-13
> > **Thanks for the clarifications**
> >
> > Thanks for your responses. My only counterpoint at this time is that the arrow next to the dictionary at each layer should not come from the blue block because it is not an output of the blue block (soft thresholding operation). Perhaps the figure should include a third block that represents the learning operation from LISTA, whose output would be the dictionary. In fact, I would say the dictionary should be an input to both the red and blue blocks.
> >
> > I am raising my score to 6.

---

### Official Review · Reviewer_2dyB · 2022-11-02

**Confidence:** 4
**Correctness:** 3
**Technical Novelty And Significance:** 3
**Empirical Novelty And Significance:** 2
**Recommendation:** 6

**Clarity, Quality, Novelty And Reproducibility:**

I don't have any difficulty in understanding the proposed method.
Also, a certain novelty can be found in the methodology itself.

**Strength And Weaknesses:**

The paper is generally interesting, and enough contribution can be recognized in the proposed approach. However, the signal recovery performance of the proposed method is worse than that of existing method (i.e., A-DLISTA) in the experimental results. Also, the organization of Sect.3 might be somewhat misleading, because it contains both the existing methods and the proposed method while it is titled "variational learning ISTA", which is the name of the proposed method.

**Summary Of The Paper:**

The paper considers the problem of sparse vector recovery with unknown dictionary and time varying sensing matrix, and proposes a data-driven approach using variational learning. The signal recovery performance of the proposed method as well as the possibility of out of distribution detection are demonstrated via numerical experiments.

**Summary Of The Review:**

The paper considers a timely and important problem, and work in the paper is solid and interesting, while the performance gain shown in the numerical results might be rather marginal.

---

> ### Author Response · Authors · 2022-11-18
> **Thank you for the suggestions**
>
> In what follows we try to address all the concerns raised by the reviewer (moreover, in the revised version of the manuscript we colored the new text, in the main body of the paper, in blue)
>
> **Q: The paper is generally interesting, ...**
>
> A: We thank the reviewer for pointing this out. A-DLISTA is not an existing model. Indeed, it is one of our contributions, although it is not the focus of the paper.  What we argue with the A-DLISTA model is that there is a necessity for a model that tries to solve the sparse recovery problem in a compressed sensing framework to be adaptive to different measurement scenarios. Such an assumption is orthogonal to most of the research in this field that assumes a fixed measurement scenario, i.e., sensing matrix. To prove the benefit of an adaptive model, we added in Table 1, 2, and 3 ablation results in which we considered the A-DLISTA without the augmentation network. We named such a model DLISTA. As we can from the results, the augmentation model plays a significant role in allowing the A-DLISTA to obtain the highest performance among other models.
> Therefore, our first contribution is to propose a model capable of adapting its parameters to such non-static scenarios. Subsequently, we try to relax the assumption about the existence of a sparsifying dictionary (or that perhaps only a single dictionary exists). Although we pay a small price in terms of performance compared to A-DLISTA, VLISTA gains the power of rejecting OOD samples which is something that none of the previous LISTA-like models can do. We motivated the reason why VLISTA has a lower performance than A-DLISTA in the revised version of the manuscript. However, if the two subsequent steps are not clear, we can restructure Sec.3 for camera ready if the paper will be accepted.
>
> **Q: The paper considers a timely ...**
>
> A: We thank the reviewer for appreciating our work. We added two new baselines and a new dataset to our previous experimental results. Moreover, we want to stress here that in terms of performance, A-DLISTA outperforms other non-Bayesian models, especially when we consider a low number of measurements. On the other hand, VLISTA gains the ability to detect OOD and its performance is utterly higher than BCS.

---

> ### Author Response · Authors · 2022-12-12
> **Addressing concerns**
>
> Dear review 2dyB,
>
> As the discussion period is closing, we would like to thank you again for the time and effort you dedicated to reviewing our work. We tried to address your concerns at our best and updated the manuscript accordingly. We are looking forward to your comments on our rebuttal.
>
> **Signal Recovery Performance:**
> We employ two steps to achieve our final formulation for the VLISTA framework. First, we propose an adaptive (non-Bayesian) strategy for sparse signal recovery (A-DLISTA), and then we move to a Bayesian approach, VLISTA, that allows us to drop any assumption on the uniqueness and existence of a ground truth dictionary.
>
> **Numerical Results:**
> Following the reviewers' suggestions, we conducted experiments on an additional dataset and added new comparisons with baselines. All the new results confirmed the conclusions we had previously drawn. Our A-DLISTA outperforms all the non-Bayeisan models while VLISTA reached better performance than BCS.

---

### Official Review · Reviewer_3LdD · 2022-11-02

**Confidence:** 4
**Correctness:** 3
**Technical Novelty And Significance:** 3
**Empirical Novelty And Significance:** 2
**Recommendation:** 6

**Clarity, Quality, Novelty And Reproducibility:**

The main ideas are clearly presented in the paper. Learning dictionaries in an adaptive measurements environment seems to be novel (though I might miss works that might have recently been published on that topic).

**Strength And Weaknesses:**

Strengths

- The paper is well-written and easy to follow. The main ideas are well explained.
- Allowing to learn the dictionary in a measurement adaptive environment is quite interesting. Also, the probabilistic formulation of the problem, which leads to VLISTA, uses ideas of Bayesian deep learning in the unrolled optimization framework, and thus it has its own interest.

Weaknesses
- I understand that assigning Gaussian distributions in the prior and posterior simplifies things a lot regarding the implementation of the algorithm. However, isn't quite restrictive in practice? I wonder if other distributions can be modeled in that framework. E.g., using hierarchical models that encode heavy-tailed distributions.
- In the experiments,  Section 4.1, it seems that A-DLISTA outperforms ISTA and LISTA. However, it is not clear what is the measurement matrix that is given as input to these algorithms. Does LISTA know that the measurement matrix changes over time?
- It is not well explained why VLISTA works so much worse than A-DLISTA.
- In the OOD experiment, there is no comparison with any other competing method. Why BSC algorithm is not included in the experiments?
- In Theorem 1, since $\theta_t,\gamma_t$ are learned by the algorithm. Hence for the sufficient condition given in (7)  to be satisfied, it is important $\theta_t$, and $\gamma_t$ converge as t ( here denotes the number of layers) grows. Can the authors provide some insight regarding the convergence of these parameters? Moreover, it would be interesting for the authors to show experimentally that this condition is satisfied at a certain number of layers T.

Minor
- There seems to be an issue with statement 1 in Theorem 1. The conclusion should be $\mathrm{supp}(z_{t}) \subseteq \mathrm{supp}(z^\ast)$.  It seems there is a typo there.


**Summary Of The Paper:**

The authors present two unrolled ISTA-type algorithms for compressed sensing. The first approach is amenable to adaptive measurements matrix while learning the sparsifying dictionary per layer. The second approach called as VLISTA places a probability distribution the dictionaries. This provides a probabilistic way to learn dictionaries which could be useful in out of distribution (OOD) detection problems.

**Summary Of The Review:**

The authors present two approaches for learning sparse representations and dictionaries under the unrolled optimization framework.
The ideas are simple but interesting. The authors provide a sufficient condition for the algorithm to converge to the correct support and provide an error bound. Experimental results demonstrate the efficiency of the proposed algorithms on simulated data and on MNIST dataset. The paper could be significantly improved if the authors could better motivate the selection of the Gaussian distributions for the priors and posteriors and provide more insight into how restrictive these assumptions are in practical settings. The experimental section could also be improved by providing experimental results on datasets such as CIFAR dataset and more extensive results of the probabilistic method (VLISTA).

-----------------------------
Post-rebuttal update:
I wanted to thank the authors for their time and effort in addressing the reviewers' comments. I am pretty satisfied with the responses, and I  appreciate the changes they made to the paper. I  thus raise my score to 6.

---

> ### Author Response · Authors · 2022-11-18
> **Thank you for the detailed comments**
>
> In what follows we try to address all the concerns raised by the reviewer (moreover, in the revised version of the manuscript we colored the new text, in the main body of the paper, in blue)
>
> **Q: I understand that assigning Gaussian distributions ...**
>
> A: Indeed, Gaussian distributions were used mostly for computational and implementation convenience. The framework is not at all restricted in that regard and, in fact, is quite flexible to support any flexible distribution family. Some examples would be scale mixtures of Gaussians (to get heavier tails) or even distributions arising from normalizing flows [1]. The main requirement for a simple learning objective is that the distributions involved are reparametrizable [2] (so that we can obtain gradients of the random samples with respect to their parameters) and we can evaluate / differentiate their density.
> [1] Rezende, D., and Shakir M.. "Variational inference with normalizing flows." International conference on machine learning. PMLR, 2015.\newline
> [2] Kingma, D. P., and Welling M.. "Auto-encoding variational bayes." 2013
>
> **Q: In the experiments, Section 4.1, ...**
>
> A: We thank the reviewer for the insightful question. Before to begin to train any model, we randomly generated a sensing matrix for each data sample. Then, while training, for each sample we evaluated its observations by multiplying the ground truth signal, e.g., an image, by the corresponding sensing matrix.  Given its formulation, LISTA does not require to have the sensing matrix as input. Therefore, such a model is “aware” of the varying sensing matrix only through a varying relation between the ground truth signal and the observations. Hence, while training, the parameters of LISTA will interpolate the relation between ground truth signal and observations across a large varierty of measurement setups, i.e., sensing matrices. Thus, obtaining non-optimal performance when a given sensing matrix is used. To better explain the procedure we use to generate the data and train the models, we report detailed descriptions in Appendix B of the revised manuscript.
>
> **Q: It is not well explained ...**
>
> A: There are few reasons that can explain why VLISTA obtains lower performance than A-DLISTA. On contribution might come from the noise that is naturally injected at training time due to the random sampling procedure to generate the dictionaries. Another contribution to lower performance is represented by the amortization gap that affects all models based on amortized variational inference. However, although VLISTA shows lower performance than A-DLISTA, it is important to notice that it still performs better than BCS. Moreover, it has the ability to detect OODs, a characteristic that Bayesian models only possess. We accordingly modified the manuscript.
>
> **Q: In the OOD experiment, ...**
>
> A: We thank the reviewer for pointing this out. We added a comparison between VLISTA and BCS in Fig. 2 of the updated manuscript. As we can see from the mentioned figure, VLISTA reaches a higher rejection power than BCS.
>
> **Q: In Theorem 1, since  ...**
>
> A: To start, note that  $\theta_t$ and $\gamma_t$ are  deterministic function of $\Phi$ and $\Psi_t$. Since for each inverse problem, the sensing matrix $\Phi$ does not change across layers, the  dynamics of $\theta_t$ and $\gamma_t$ depends on how the learned $\Psi_t$ changes with $t$. If the learned $\Psi_t$'s converge to a fixed dictionary for a large $t$, then $\theta_t$ and $\gamma_t$, trivially, converge to a fixed value as they are a  function of  $\Phi$ and $\Psi_t$. However, two important caveats are in order. First, even if we choose a very large number of iterations $t$ (i.e., large number of layers), it is not clear if $\Psi_t$ needs to converge to a fixed dictionary after training. Second, the main idea behind learned iterative algorithms is to avoid large number of iterations. In our case, we tend to get good results with only three iterations, for which the learned $\Psi_t$ are different across layers. Finally, we would like to emphasize that there is a difference between convergence of the learned algorithm, which is about getting progressively smaller reconstruction error, and the convergence of learned parameters as we increase the number of layers. The former notion is central to our task.
>
> **Q: There seems to be an issue ...**
>
> A: We would like to thank the reviewer for spotting this typo. We have fixed it in the main paper.
>
> **Q: The authors present two approaches for learning ...**
>
> A: We thank the reviewer for his suggestions. Concerning the choice of Gaussian distribution to model the prior and posterior over dictionaries, we refer the reviewer to the answer we gave to his first question.
> Concerning the datasets, we added results on the CIFAR10 dataset in Table 2. The newly obtained results on the CIFAR10 dataset, confirm the conclusions we draw from the previous datasets we used.

---

> ### Author Response · Authors · 2022-12-12
> **Addressing concerns**
>
> Dear review 3LdD,
>
> As the discussion period is closing, we would like to thank you again for the time and effort you dedicated to reviewing our work. We tried to address your concerns at our best and updated the manuscript accordingly.
> We are looking forward to your comments on our rebuttal.
>
> **On Gaussian distribution:**
> Although we used Gaussians to model the dictionary distribution, our framework is not at all restricted in that regard. Indeed, it is quite flexible to support any flexible distribution family such as mixtures of Gaussians (to get heavier tails) or even distributions arising from normalizing flows.
>
> **OOD experiment:**
> Following the reviewer’s suggestion, we added a comparison between BCS and our model. Concerning the OOD detection experiment, we notice that VLISTA reaches higher rejection power than BCS fordifferent levels of noise.

---

### Author Response · Authors · 2022-11-29
**Thank for the reviews**

We would like to thank the reviewers for helping us improving our paper.
We tried our best to address all the reviewers' concerns by adding more experiments, comparing with other baselines and fixing typos.
To ease the reviewers, we highlighted in blue the updated parts of the paper. Moreover, we extended the appendices to include more details that will help the reproducibility of our results.

---

### Decision · Program_Chairs · 2023-01-20

**Decision:**

Reject

**Justification For Why Not Higher Score:**

Although the exposition in this paper has been improved via the revision and the added experimental results are fine in demonstrating efficiency of the proposed methods, I feel that there are still several points that require further clarification, as detailed above.

**Justification For Why Not Lower Score:**

N/A

**Metareview: Summary, Strengths And Weaknesses:**

This paper proposes a variational-learning-based ISTA for solving compressed sensing problems with an unknown dictionary. As an intermediate, it also proposes A-DLISTA which, unlike DLISTA (3) that learns $\theta_t,\gamma_t,\Psi_t$ in an end-to-end manner, uses an augmentation network that outputs $\theta_t,\gamma_t$ from $\Phi,\Psi_t$ and is trained in an end-to-end manner, in order to allow $\theta_t,\gamma_t$ to be determined adaptively rather than being fixed as a result of the end-to-end training. The main proposal, VLISTA, is built on A-DLISTA by further adding to it a prior model for the dictionaries $\Psi_{1:T}$ and a set of variational distributions to be optimized by maximizing the ELBO (10). The setting discussed in this paper where the sensing matrix $\Phi$ is known but varies from sample to sample and where the sparsifying dictionary $\Psi_o$ is unknown and to be estimated would appear in applications such as MRI reconstruction and MIMO channel estimation. The proposed methods were shown to perform well in such a scenario.

Of the three reviewers, one initially rated well below the acceptance threshold, and it appears that it is mainly due to rather unclear exposition of the original manuscript and limited experimental comparison with baselines, which constituted the main weaknesses of the initial version. Upon reading the revised version of this paper by myself, I think that clarity of the presentation is not enough even after the revision. For example, regarding the ELBO objective (10), the authors used an undefined symbol $x_{gt}$ (I guess that it is for "$x$ ground truth" or something like that), and considered the likelihood given $\hat x_t=x_{gt}$ for all $t\in[1:T]$ in the first term of the ELBO, which would need further justification as it should be inconsistent with their use of not $x_{gt}$ but $\hat x_{t-1}$ in the second term of the ELBO. I would like to mention that on the right-hand side of (9) $\hat x_{1:T}$ should appear whereas it does not.

I would also like to add that a more detailed explanation on why different dictionaries $\Psi_{1:T}$ and their distributions are assumed in the proposal of VLISTA, from the viewpoint of Bayesian inference. Although from the deep-unfolding viewpoint it would be seen natural to consider different dictionaries for different layers, one can argue that it would conceptually be more natural to assume a single dictionary $\Psi$ as well as a single prior for it, independent of $t$.